# Vaccination Week in the Americas: An Ongoing Initiative to Strengthen and Sustain Measles and Rubella Elimination in the Region

**DOI:** 10.3390/vaccines12070812

**Published:** 2024-07-22

**Authors:** Alba Maria Ropero, Hannah Kurtis, Lauren Vulanovic, Pamela Bravo-Alcántara, Maite Vera Antelo, Margherita Ghiselli

**Affiliations:** Pan American Health Organization, Washington, DC 20037, USA; roperoal@paho.org (A.M.R.); kurtisha@paho.org (H.K.); brownlau@paho.org (L.V.); bravopam@paho.org (P.B.-A.); veraanmai@paho.org (M.V.A.)

**Keywords:** measles, rubella, elimination, immunity gap, Vaccination Week in the Americas

## Abstract

Vaccination Week in the Americas (VWA) is a yearly regional initiative that promotes the benefits of vaccination to all persons in the region. In its 22-year history, more than 1.15 billion people have been reached under the framework of VWA across more than 40 countries and territories. This review examines multiple PAHO and WHO data points, documents and reports related to measles/rubella vaccination coverage and VWA since its inception. Its goal is to document the impact that the VWA has had in maintaining and accelerating measles and rubella disease elimination, in the context of PAHO’s Disease Elimination Initiative. The results suggest that VWA’s contributions to measles and rubella elimination have been substantial. Every year, VWA promotes (a) renewed political commitment to the immunization program from the highest political authorities of Member States; (b) vaccination operations to close immunity gaps, recover under-vaccinated persons, and reach chronically underserved populations; and (c) the dissemination of messages on the benefits of vaccination through regional and national communications campaigns. VWA will continue to be an important contributor to disease elimination efforts in the Americas, even as new targets are set in response to the evolving epidemiological landscape.

## 1. Introduction

Vaccination Week in the Americas (VWA) is the Americas’ yearly initiative for the promotion of the regional immunization program and the closure of the most urgent immunity gaps. Launched in 2003, it serves as a coordinated and flexible platform for countries to offer vaccines and promote the benefits of immunization to health workers and the public. Over the years, countries have taken advantage of this event to complement the operations of the routine immunization program or launch mass vaccination campaigns and mop-up operations. This review documents the extensive contributions of the VWA to achieve and sustain the elimination of measles, rubella and congenital rubella syndrome (CRS) in the Americas.

This review considered the results of the VWA as reported in each year’s final report for the event, as well as the yearly data presented in the reports to the Regional Measles-Rubella Elimination Monitoring and Re-verification Commission, the PAHO electronic joint reporting form (eJRF) and the WHO/UNICEF Estimates of National Immunization Coverage (WUENIC). This review aims to describe the contributions of the VWA in maintaining and accelerating measles and rubella disease elimination, in the context of PAHO’s Elimination Initiative [1]. We conclude by presenting some limitations of the platform, as well as lessons learned to further the goal of measles/rubella disease elimination in the Americas.

## 2. History of the Vaccination Week in the Americas

Vaccination Week in the Americas (VWA) is an annual initiative that has been celebrated in the Americas for the last 22 years. As reported previously [2,3,4], an outbreak of measles linked to endemic transmission emerged along the border between Venezuela and Colombia in 2002. In response to this crisis, the ministers of health of the Andean Region and then-Director of the Pan American Health Organization (PAHO) Dr. George A. O. Alleyne signed the Sucre Agreement, which specified several actions to prevent future outbreaks, including the establishment of a coordinated, multi-country vaccination week among Andean countries.

The first VWA was celebrated in June 2003, with the participation of 19 countries and territories across the Americas, resulting in the vaccination of more than 16 million people [4]. In September 2003, during PAHO’s 44th Directing Council, health ministers endorsed Resolution CD44.R1 “Sustaining Immunization Programs-Elimination of Rubella and Congenital Rubella Syndrome (CRS)”, which included language supporting “the implementation of an annual hemispheric Vaccination Week […] targeting high-risk population groups and underserved areas.” [5]. This resolution was significant since it provided the political mandate for countries’ participation in VWA in the years to come.

Also, the VWA served as an inspiration for the creation of World Immunization Week (WIW) by the World Health Organization (WHO). Following the example of the Americas, other WHO regions created their own regional vaccination weeks (EURO in 2005; EMRO in 2010; AFRO in 2011; WPRO in 2011; and SEARO in 2012). During the 56th World Health Assembly in May 2012, Resolution WHA 65.18. called on “Member States to designate the last week of April, when appropriate, as World Immunization Week”, with the aim to prioritize and fully fund immunization programs, ensure access to vaccines for individuals throughout the life course, encourage vaccine-related innovation, and inspire vaccine advocacy efforts [6].

In its 22-year history, more than 1.15 billion people have been vaccinated under the framework of VWA. Of note, this figure includes the administration of at least 78.8 million doses of measles and rubella-containing vaccine (MMR or MR), of which more than 40.5 million were adult doses.

## 3. Contributions to the Measles/Rubella Global Elimination Initiative

### 3.1. A Platform to Accelerate PAHO’s Elimination Initiative

Since its inception, the VWA has included several elements that actively promote, accelerate and maintain the target of vaccine-preventable disease (VPD) control and measles/rubella elimination in the Americas. These include the following:The annual opportunity to revitalize political and financial commitment towards sustaining VPD control and measles/rubella elimination across the region. (a)Targeted vaccination operations designed to close the most urgent immunity gaps and reduce the pool of susceptible persons.(b)Flexibility to adapt the VWA platform to the national context and address the most urgent immunity gaps, in accordance with the country’s epidemiological situation and public health priorities. This is often done through outreach activities, mass vaccination campaigns and mop-up operations.(c)Offer of equitable access to vaccination services for remote and chronically underserved populations.Cooperation between countries to promote shared health goals such as measles/rubella elimination, including joint cross-border vaccination operations between Member States.Widespread dissemination of messages and communication materials at both the country and regional level, which promote the benefits of vaccines to health workers and the public.

Furthermore, in 2019, the PAHO Member States approved the Resolution PAHO Disease Elimination Initiative during the 57th Directing Council. This policy provides a common and sustainable framework with prioritized lines of action to orient and guide countries of the region as they work toward the elimination of a group of 30 priority communicable diseases—including measles and rubella—and related conditions. The framework is strategic, inclusive, standardized, and multisectoral, and can be adopted, adapted, and implemented in a staged manner by the countries of the region according to their national and local contexts and priorities [1,7]. Given its flexibility and wide, sustained implementation across the region, the VWA was quickly included among the strategies that countries implement to advance this Initiative.

As with the global switch from trivalent to bivalent oral polio vaccine (OPV) in 2016 and the introduction of the COVID-19 vaccine in 2021, Member States can now use the VWA to accelerate the implementation of this Initiative in their own territories. Its regular implementation allows all countries to prepare their advocacy and vaccination activities in a timely fashion, as well as to tailor interventions and coordinate cross-border operations where vaccination coverage rates are weakest.

### 3.2. Political Priority on the National Agenda of Member States

One of the main strategic objectives of the VWA is to maintain the topic of immunization at the forefront of the political agenda of Member States, partners and donors.

At the regional level, the yearly launching ceremony of the VWA is hosted by the PAHO Director and represents an opportunity to engage directly the Ministries of Health, civil societies and partners of the region and advocate in favor of a strong, well-financed regional immunization program. Each year, Ministers of Health of select countries and representatives of strategic partnerships are invited to share messages in support of immunization programs, so the Director’s message is amplified across the region. Also, the ceremony serves as a call to action to governments to increase vaccination coverage rates by investing in all aspects of the national immunization program, including human resource capacity and integration of vaccination services into the primary healthcare system.

These same objectives are replicated in each country through the national-level VWA launching ceremonies. Here, governments, civil societies and their partners publicly renew their commitment to the national immunization program. In the months leading up to VWA, the leadership of the immunization program and the staff of the PAHO Country Offices engage national authorities at the highest levels to ensure their vocal support and advocacy in favor of vaccination operations and programs. The successful participation of heads of state, their spouses and ministers of health has been a key outcome of these events. On occasion, political commitment to national immunization programs has been showcased through the creation of ministerial declarations, which are signed during a VWA launching ceremony to bring maximum visibility to the event. A recent example occurred during VWA 2023, when the countries of the Caribbean agreed to strengthen national immunization programs through the Declaration of Nassau [8]. These statements often attract opportunities for cooperation and collaboration between governments, partners and the private sector, which may translate into financial or in-kind support towards national vaccination activities.

Finally, the PAHO Director holds a press conference the week prior to the VWA to discuss vaccines and the benefits of vaccination with the wider public, as well as to address the most urgent questions related to the regional immunization program.

### 3.3. Vaccination Operations

The VWA is designed to be flexible in how it is implemented country-to-country and year-to-year. Countries set their own goals within the regional strategic framework, so the initiative becomes an annual opportunity to advance public health priorities and close the most urgent gaps within the performance of the national EPI. Below we describe some of the strategies that have been implemented under the auspices of the VWA to bring vaccination services closer to the public and close the most urgent immunity gaps.

#### 3.3.1. Outreach Efforts

Vaccination outreach efforts are one of the hallmark activities carried out by countries during the celebration of the VWA. In the context of VPD control and measles/rubella elimination, this has taken the form of special efforts to ensure that isolated or vulnerable populations are offered vaccination services at convenient times and places. For example, countries such as Argentina, Brazil, Bolivia, Colombia, Ecuador, Nicaragua, Paraguay, Peru and Venezuela have used VWA to vaccinate indigenous groups against measles and rubella, with outreach teams often traveling by boat, plane and foot to reach the most remote villages.

Another essential contribution of VWA to VPD control and measles/rubella elimination is cross-border vaccination activities. For example, Honduras, Guatemala, Nicaragua and Panama carried out synchronized measles follow-up vaccination campaigns as part of their VWA celebrations in 2008. These campaigns were combined with a series of consecutive VWA launching ceremonies organized in border areas between the United States and Mexico and across Central America around the theme of a “Caravan for Health” and the shared slogan “Get on Board, Get Vaccinated”. Common themes mentioned during these ceremonies included the need for international cooperation and solidarity to prevent VPD spread. Finally, in 2024, cross-border vaccination activities were organized between Mexico, Belize and Guatemala; between Brazil and Colombia; and between Brazil and Paraguay.

Also, countries have used the VWA to reach individuals whose profession puts them at higher risk for measles/rubella exposure, compared to the general population. Many countries—including Bolivia, Grenada, Jamaica, Panama, St. Kitts and Nevis and Trinidad and Tobago—used the VWA to intensify their vaccination activities against measles and rubella among health workers. Countries with a high influx of tourists (ex., Caribbean territories, the Dominican Republic, Mexico) targeted professionals who are in close contact with travelers, including taxi drivers and baggage handlers. For example, Anguilla vaccinated immigration, police and customs officers, while the Dominican Republic vaccinated individuals working in the tourism industry. Peru vaccinated airport workers and hotel staff.

Countries have used the VWA platform to make the vaccination setting more friendly for the public. The province of Cordoba in Argentina hosts the annual *La Noche de las Vacunas* (“Vaccination Night”), where local hospitals open their doors around 8:00 pm for an all-night party where thousands of people of all ages can play games, listen to live music, watch performances, have dinner and get vaccinated [9]. Countries like Belize, Bolivia, Costa Rica, Dominican Republic, Jamaica and Trinidad and Tobago completed MR vaccination operations in schools and daycare centers, while countries like Argentina, Brazil and Trinidad and Tobago set up mobile routine vaccination services at town squares, concert venues, shopping malls, metro stations, senior centers, banks and other public spaces. Intensified efforts to peri urban areas are hallmarks of the VWA, in its effort to reach chronically underserved populations.

Finally, migrant populations have benefited from VWA events. For example, the Ministry of Health of the Dominican Republic has worked with non-governmental organizations and community groups to reach migrant populations who have limited or no access to routine health services within the country, but whose vulnerable situations places them at high risk for contracting VPDs if they remain susceptible. Also, Trinidad and Tobago used the VWA to expand vaccination service hours and make routine immunization more accessible to all persons, including migrant populations. The countries of St. Kitts and Nevis and Turks and Caicos implemented outreach operations in migrant communities to encourage vaccination for all persons.

#### 3.3.2. Mass Vaccination Campaigns and Mop-Up Operations against Measles/Rubella

Current PAHO recommendations state that the measles-containing vaccine should be administered at 12 and at 18 months of age [10]. With regional vaccination coverage rates fluctuating between 85% and 91% from 2017 to 2022 [11]—well below the 95% target set for the region—an accumulation of non-immune children has resulted over time [11]. This includes both children who were never vaccinated and those who were vaccinated but failed to respond to the vaccine. With each successive birth cohort, the number of children susceptible to measles increases. This build-up is the most serious obstacle to measles elimination. High vaccination coverage through routine health services is essential, yet alone it is clearly not sufficient to reach and maintain measles elimination.

Mass vaccination campaigns are a rapid and resource-intensive strategy to close the immunity gaps left by the routine immunization program, quickly reduce the number of susceptible persons and maintain the elimination targets that the Americas already achieved. As part of the VWA’s 22-year history, 25 high-quality “follow-up” campaigns were implemented at least 14 countries implemented high-quality “follow-up” mass campaigns against measles/rubella among children younger than five years, with the purpose of providing a second vaccination opportunity to the entire cohort. Through these campaigns, close to 49 million children were vaccinated with measles-containing vaccines. The most frequent targeted group was children 1–4 years old (52%), while 28% of the campaigns targeted older children (up to 14 years). Administrative coverage rates ranged from 73% to 114%, with a median of 96% (Table 1).

During this same period, six countries used the VWA platform to implement “speed-up” campaigns, a one-time mass intervention targeting adolescents and adults aged 25–39 years to eliminate rubella and CRS, as well as advance towards measles elimination. In these campaigns, close to 45 million individuals were vaccinated with the MR vaccine. The coverage rate achieved ranged from 98% to 99%, with a median of 99% (Table 2). Also, multiple countries and territories in the Caribbean carried out an intensification of vaccination activities to target different age groups and ensure that all receive one or two doses of the MMR vaccine. This outreach strategy contributes to achieving high and homogeneous population immunity rates.

With the support of PAHO, countries of Latin America implemented synchronized speed-up campaigns in a short timespan: 89% of these campaigns were conducted between 2004–2008, while 38% were launched or implemented during VWA.

#### 3.3.3. Response Operations against the 2017 and 2018 Outbreaks in Venezuela and Brazil

In July 2017, Venezuela reported a measles outbreak in the southern state of Bolivar. Since the country was not able to close the outbreak within 12 months from the notification of the first confirmed case, it was reclassified as having reestablished endemic measles virus transmission. As a result, the whole region lost the measles-free status it had achieved in 2016. By February 2018, the virus spread to Roraima state in Brazil, which borders Bolivar. By 2019, Venezuela had reported 7054 confirmed cases: 727 in 2017; 5799 in 2018; and 1722 in 2019.

During the VWA in April 2018, Venezuela implemented an outbreak response intervention by intensifying MR vaccination activities among children between ages 6 months to 15 years, with a focus in the nine states with the highest proportion of confirmed cases and highest cumulative incidence rates. Additionally, this campaign was an opportunity to offer the tetanus and adult diphtheria (Td) vaccine to persons ages 7 to 15 years to close the ongoing diphtheria outbreak. The mass vaccination campaign was eventually expanded to all 24 states of Venezuela. Between April 2018 and May 2019, 8.9 million children received the MR vaccine, for a national coverage rate of 104%.

By April 2018, only 3 of the 27 states of Brazil had been affected by the measles outbreak. Therefore, the country selected to implement its follow-up MR vaccination campaign in August 2018, outside the framework of the VWA, to allow for additional time to implement micro-planning exercises as well as complete large procurement and distribution operations for vaccines and supplies. In the three affected states, the VWA focused its MR vaccination activities among migrant populations found in shelters or at border posts.

#### 3.3.4. Communications and Community Engagement

Each year, PAHO develops a regional VWA communication campaign. Materials include a slogan that reflects current events and priorities; digital and print materials (including posters, banners, stickers, gifs, videos, and images), public service announcement (PSA) infographics, and a social media package with content tailored for a variety of platforms. Countries can adapt these materials to meet the goals and targets of their own VWA campaign and ensure that all images and messages reflect the ethnic and linguistic composition of their populations. Over the years, many countries adapted these materials and slogans to promote the mass vaccination campaign they implement against measles and rubella.

While the option for adaptation remains available to countries, there have been several years where the regional communication campaign materials included a specific call to action centered around measles and rubella elimination activities. This concentration of efforts around a specific topic has occurred mostly in years when the Americas have hosted international sporting events or have contributed a significant number of players to the game. For example, in 2014 and 2018—both FIFA World Cup years—the VWA regional communication campaign themes used soccer slogans and graphics to focus on strengthening surveillance operations and vaccination activities to prevent measles and rubella outbreaks. For the 2007 Cricket World Cup—hosted in the West Indies—Caribbean countries used the VWA to bolster their epidemiological surveillance operations to identify any suspected cases of measles or rubella among participants or spectators. At the same time, communication campaigns in the hosting countries emphasized the importance of vaccination to prevent the spread of measles and rubella.

Community engagement is another critical element of VWA communications campaigns. Over the years, countries developed multiple approaches to engage with different audiences at the local level in order to promote vaccination against measles/rubella. For example, collaboration with church leaders has allowed the VWA to promote key vaccination messages among their congregations. In other cases, high-level health professionals or political leaders have been invited to speak about vaccination opportunities during religious services. Often, churches serve as the location for health fairs and MR vaccination during the VWA week.

A pivotal moment was the introduction of COVID-19 vaccines in the region of the Americas in January 2021. At this time of uncertainly and high pressure for rapid vaccine introduction, the novelty of the vaccine itself raised considerable doubts about the vaccines’ safety and effectiveness. In multiple countries, these doubts threatened to spread to antigens of the routine immunization program, such as measles-containing vaccines. PAHO used the VWA as an opportunity to engage with the public, health workers and local opinion leaders, answer questions and concerns, address rumors and debunk myths. The communication materials of the VWA were designed to provide consistent and up-to-date answers to the most frequent questions around the region.

## 4. Lessons Learned

### 4.1. Political Priority

The main lesson learned from the VWA is the importance of a periodic, well-coordinated and flexible opportunity for governments and heads of state to address the importance of vaccines and vaccination operations, regardless of other political priorities or events. Indeed, the VWA often served as an opportunity to showcase how vaccines can alleviate the outcomes of a public health emergency. This yearly appointment allows all Member States to remind their populations of the risk of vaccine-preventable diseases and of the importance of routine vaccines for the health and well-being of all communities.

Another lesson learned is the convenience of the VWA as a well-established platform to facilitate and accelerate the implementation of region-wide public health interventions. As with the trivalent–bivalent OPV switch in 2016 and the introduction of the COVID-19 vaccines in 2021, the VWA can ease these moments of transition by presenting them within a familiar and reliable framework to both the health workforce and the public.

### 4.2. Vaccination Operations

The VWA provides a platform to implement synchronized and massive vaccination campaigns, thus increasing population immunity levels quickly. Each year, the VWA underscores the need to “reach the unreached” by focusing efforts to bring vaccines to vulnerable and chronically underserved populations. This effort is a cornerstone of the elimination strategy of measles and rubella.

Flexibility is another attribute of the VWA platform. Countries set their own vaccination goals, so the initiative becomes an annual opportunity to advance national public health priorities. At the same time, by maintaining measles and rubella elimination as a regional target, PAHO provides ongoing guidance and support to countries and partners to achieve this common goal while allowing space for Member States to implement the vaccination tactics that work best for their context.

### 4.3. Communications and Community Engagement

In the area of communications, there are two main lessons learned. The first is the importance of allowing countries to adapt the VWA communications materials (both print and digital media). Regional VWA materials already reflect the ethnic diversity of the Americas, so countries have multiple options to select the images that best reflect their populations. Also, all information materials developed by the regional level (for example, talking points on measles/rubella surveillance and vaccination activities, MR coverage rates, number of suspected and confirmed cases) are designed so that countries can easily replace region-level data with their own national and local information and can further tailor the materials to speak to their own priorities.

The second lesson learned is to use the VWA platform to increase the public’s risk perception of measles and rubella disease while decreasing the risk perception associated with the safety and effectiveness of MR vaccination. This can be done by emphasizing the risk of severe illness, hospitalization, encephalitis, pneumonia, birth defects, miscarriage and death—all of which can be prevented through vaccination. Also, communication materials have been used to highlight the safety and effectiveness of the measles-containing vaccines, which have been consistently proven over decades of use in the general population. The VWA provides multiple opportunities for public health experts to promote these points, as well as to meaningfully engage with health workers and the public on MR vaccination, address doubts and provide reliable and updated information.

### 4.4. Limitations

Upon review of the VWA documentation, we notice two main limitations to its contributions of the measles and rubella disease elimination initiative. First, since the VWA is always celebrated the last week of April, its operations cannot be timed to address public health emergencies. An example is the measles outbreak that was declared between Venezuela and Brazil in July 2017. Both affected and neighboring countries had to implement rapid response operations outside the context of the VWA to prevent further transmission. Also, the fact that the VWA serves as both an outreach activity to vaccinate zero-dose children and as a mop-up operation for under-vaccinated persons means that the target population for a specific antigen is often larger than the one specified for routine immunization activities. In the case of MR vaccination, the mass vaccination follow-up campaigns organized for the VWA often reached all children younger than five years, while the national immunization program recommends vaccinating children at 12 and at 18 months of age. Therefore, we cannot ascertain that VWA vaccination operations closed the immunity gaps left by the routine immunization programs of Venezuela and Brazil. Nonetheless, we presented strong evidence that VWA operations in both affected and neighboring countries contributed to shrinking the pool of persons susceptible to measles and rubella in the Americas—thus possibly preventing additional outbreaks.

A second limitation is the arrival of the COVID-19 pandemic to the Americas in 2020. While most VWA events at the national and regional level continued uninterrupted, Colombia had to extend its mass campaign vaccination operations throughout 2020 to achieve its MR coverage targets. Also, Honduras, Bolivia and Paraguay postponed their MR follow-up vaccination campaigns originally scheduled for 2021. Furthermore, since the 2020 VWA was limited in scope and the 2021 VWA was dedicated mostly to COVID-19 vaccine introduction, the countries of the Americas have had limited opportunity to date to employ this platform as an extension of the PAHO Elimination Initiative. Future iterations of the VWA will likely be more explicit in promoting the platform for this purpose.

## 5. Conclusions

For the last 22 years, the countries and territories of the Americas have used the VWA to renew their political and financial commitment to the regional immunization program, advance in the elimination of vaccine-preventable diseases—including measles and rubella—and remind the public of the benefits of vaccines for the health and well-being of all communities. Its consistent implementation during the last week of April allowed countries to join this yearly appointment through coordinated operations and messages and offers the public a familiar interface with the national vaccination program. In its next iterations, the VWA will continue to support the PAHO Elimination Initiative while it contributes to closing the most urgent immunity gaps.

## Figures and Tables

**Table 1 vaccines-12-00812-t001:** Follow-up mass vaccination campaigns carried out during Vaccination Week of the Americas, 2003–2023.

	First Campaign				Second Campaign				Third Campaign				Fourth Campaign			
Countries	Year	Age Group	Coverage (%)	Vaccinated	Year	Age Group	Coverage (%)	Vaccinated	Year	Age Group	Coverage (%)	Vaccinated	Year	Age Group	Coverage (%)	Vaccinated
Bolivia	2003	1–4 yo	95	894,634												
Paraguay	2003	1–4 yo	93	551,993	2014	1–5 yo	73	738,619								
Honduras	2004	1–4 yo	94	714,207	2008	1–4 yo	97	687,966	2012	2–4 yo	84	710,166				
Nicaragua	2008	1–4 yo	95	375,781	2012	1–4 yo	107	559,985	2016	1–4 yo	101	541,836	2022	1–6 yo	114	823,127
Venezuela	2006	1–4 yo	99	2,673,972	2014	1–5 yo	99	2,466,543								
Guatemala	2008	1–6 yo	95	1,992,185												
Panama	2008	1–4 yo	94	230,116	2018	1–4 yo	98	288,274								
Dominican Republic	2010	1–8 yo	96	1,532,705	2015	1–4 yo	95	742,792	2022	1–5 yo	95	934,329				
Costa Rica	2011	2–9 yo	93	585,121												
Ecuador	2012	6 m–14 yo	101	4,700,526												
Haiti	2012	9 m–9 yo	99	2,963,911	2016	1–4 yo	99	1,279,526								
Colombia	2021	1–10 yo	95	7,752,514												
Mexico	2021	1–9 yo	96	11,202,785												
Ecuador	2023	1–12 yo	98	3,119,470												
TOTAL				39,289,920				6,763,705				2,186,331				823,127

Source: electronic Joint Reporting Form (eJRF), World Health Organization.

**Table 2 vaccines-12-00812-t002:** Speed-up mass vaccination campaigns carried out during Vaccination Week of the Americas.

Countries	Year	Age Group	Coverage (%)	Vaccinated
Ecuador	2004	16–39 yo	98	4,982,760
El Salvador	2004	15–39 yo	98	2,796,391
Paraguay	2005	5–39 yo	99	3,753,392
Bolivia	2006	15–39 yo	98	4,015,554
Guatemala	2007	9–39 yo	99	7,172,847
Mexico	2008	19–29 yo	99	22,231,820
Total individuals vaccinated				44,952,764

Source: electronic Joint Reporting Form (eJRF), World Health Organization.

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
