# Peer review of "Vaccination Week in the Americas: An Ongoing Initiative to Strengthen and Sustain Measles and Rubella Elimination in the Region"

_vaccines, 2024, doi:10.3390/vaccines12070812_

Round 1

Reviewer 1 Report

Comments and Suggestions for Authors

Congratulations to authors! The article is timely, relevant, with deep meaning. An excellent example for other continents, and in many ways for the European continent too. I was especially inspired by the description of such practices as cross-border vaccination activities, vaccination of unusual targeted professionals who are close to travelers (taxi drivers, baggage handlers, immigration, police, customs officers, airport workers and hotel staff, initiative "Vaccination night" and many others, which must necessarily be replicated in other regions (where possible in accordance with the national, racial-ethnic, cultural characteristics of the population). Experience in influencing decision makers is especially valuable (invitation for the them to speak about vaccination opportunities during religious services. Special bravo - churches as the location for health fairs, including vaccination.  Overall - positive assessment of the article.

Author Response

Comments 1: Congratulations to authors! The article is timely, relevant, with deep meaning. An excellent example for other continents, and in many ways for the European continent too. I was especially inspired by the description of such practices as cross-border vaccination activities, vaccination of unusual targeted professionals who are close to travelers (taxi drivers, baggage handlers, immigration, police, customs officers, airport workers and hotel staff, initiative "Vaccination night" and many others, which must necessarily be replicated in other regions (where possible in accordance with the national, racial-ethnic, cultural characteristics of the population). Experience in influencing decision makers is especially valuable (invitation for the them to speak about vaccination opportunities during religious services. Special bravo - churches as the location for health fairs, including vaccination.  Overall - positive assessment of the article.

Response 1: Thank you very much for taking the time to review this manuscript, and thank you for your positive comments. Since no changes were proposed, we did not modify the manuscript from its original submission. Thank you again.

Reviewer 2 Report

Comments and Suggestions for Authors

Thank you for allowing me to review the manuscript entitled “Vaccination Week in the Americas: An ongoing initiative to strengthen and sustain measles and rubella elimination in the Region.”  The manuscript was well-written and informative. I only had a few questions.

1.        In section 3.1 A platform to accelerate PAHO’s Elimination Initiative, under 1.a) they note “Targeted vaccination operations designed to promote:”  and then goes to “b).”  Not sure what it was designed to promote.

2.       Further down, in the paragraph right after 3.1.3, they discuss the Elimination Initiative and that it is “a policy for an integrated sustainable approach to communicable diseases in the Americas . . .  committed to eliminating more than 30 communicable diseases . . . by 2030” but it seems rather vague.  Is there any further information on more specifics of the policies and its “commitment” or is the policy more a statement without substance? 

3.       In section 3.2, third paragraph, the authors note participation of “heads of state, First Ladies, and ministers…”  Are all heads of state male?  Are there no female Heads of State and their male spouse participating?  This may seem inconsequential, but perhaps it can be stated otherwise.  Not necessary, but just a thought.

4.       In section 3.3.2 the authors note coverage rates up to 114%.  How does one get a vaccine coverage rate more than 100%.  This is also noted in the table 1 (rates of 101 – 114) and in paragraph 2 of section 3.3.3.  It would also be helpful to show what the rates of immunization/coverage were before the campaign and then after.  Also might be helpful to show disease rates before the campaign and then shortly after.

5.       Section 4.2, paragraph 1 typo – in first line, change “do” to “to” and in second paragraph, last line, change “bets” to best.”

Author Response

Comments 2: 

Thank you for allowing me to review the manuscript entitled “Vaccination Week in the Americas: An ongoing initiative to strengthen and sustain measles and rubella elimination in the Region.”  The manuscript was well-written and informative. I only had a few questions.

  1.  In section 3.1 A platform to accelerate PAHO’s Elimination Initiative, under 1.a) they note “Targeted vaccination operations designed to promote:”  and then goes to “b).”  Not sure what it was designed to promote.
  2. Further down, in the paragraph right after 3.1.3, they discuss the Elimination Initiative and that it is “a policy for an integrated sustainable approach to communicable diseases in the Americas . . .  committed to eliminating more than 30 communicable diseases . . . by 2030” but it seems rather vague.  Is there any further information on more specifics of the policies and its “commitment” or is the policy more a statement without substance? 
  3. In section 3.2, third paragraph, the authors note participation of “heads of state, First Ladies, and ministers…”  Are all heads of state male?  Are there no female Heads of State and their male spouse participating?  This may seem inconsequential, but perhaps it can be stated otherwise.  Not necessary, but just a thought.
  4. In section 3.3.2 the authors note coverage rates up to 114%.  How does one get a vaccine coverage rate more than 100%.  This is also noted in the table 1 (rates of 101 – 114) and in paragraph 2 of section 3.3.3.  It would also be helpful to show what the rates of immunization/coverage were before the campaign and then after.  Also might be helpful to show disease rates before the campaign and then shortly after.
  5. Section 4.2, paragraph 1 typo – in first line, change “do” to “to” and in second paragraph, last line, change “bets” to best.”

Response 2: Thank you for your time in reviewing this manuscript. We implemented all recommended changes. Specifically:

  1. The sentence was completed, to read: "Targeted vaccination operations designed to close the most urgent immunity gaps and reduce the pool of susceptible persons."

  1. The paragraph was expanded, to read: "Furthermore, in 2019, the PAHO Member States approved Resolution PAHO Disease Elimination Initiative during the 57th Directing Council. This policy provides a common and sustainable framework with prioritized lines of action to orient and guide countries of the Region as they work toward the elimination of a group of 30 priority communicable diseases – including measles and rubella – and related conditions. The framework is strategic, inclusive, standardized, and multisectoral, and can be adopted, adapted, and implemented in a staged manner by the countries of the Region according to their national and local contexts and priorities [1, 11]. Given its flexibility and wide, sustained implementation across the Region, the VWA was quickly included among the strategies that countries implement to advance this Initiative."

  1. The term "First Ladies" was changed to "their spouses".

  1. Administrative coverage rates - as reported in this paragraph - use official census estimates as the denominator. It is possible that the country will vaccinate more persons than estimated because there are non-official residents within the territory and/or because the census estimates are outdated. In these cases, the coverage rate will exceed 100%. Starting in 2021, countries in the Americas started calculating their official vaccination coverage rates to include doses administered during both routine vaccination activities and mass vaccination campaigns. Before then, only doses of the routine program were considered. Therefore, this manuscript cannot account for the impact that VWA activities have had on the official vaccination coverage rate of countries before 2021. Finally, regarding disease rates before and after the campaigns, the number of confirmed measles and rubella cases remained 0 in all countries. The objective of these campaigns was to maintain the elimination targets - not respond to outbreaks. 

  1. Corrections made.